# Recent Progress on Graphene-Based Nanocomposites for Electrochemical Sodium-Ion Storage

**DOI:** 10.3390/nano12162837

**Published:** 2022-08-18

**Authors:** Mai Li, Kailan Zhu, Hanxue Zhao, Zheyi Meng

**Affiliations:** 1College of Science, Donghua University, Shanghai 201620, China; 2State Key Laboratory for Modification of Chemical Fibers and Polymer Materials, College of Materials Science, Donghua University, Shanghai 201620, China

**Keywords:** graphene, nanocomposites, sodium-ion battery, electrochemical analysis, energy storage mechanism

## Abstract

In advancing battery technologies, primary attention is paid to developing and optimizing low-cost electrode materials capable of fast reversible ion insertion and extraction with good cycling ability. Sodium-ion batteries stand out due to their inexpensive price and comparable operating principle to lithium-ion batteries. To achieve this target, various graphene-based nanocomposites fabricate strategies have been proposed to help realize the nanostructured electrode for high electrochemical performance sodium-ion batteries. In this review, the graphene-based nanocomposites were introduced according to the following main categories: graphene surface modification and doping, three-dimensional structured graphene, graphene coated on the surface of active materials, and the intercalation layer stacked graphene. Through one or more of the above strategies, graphene is compounded with active substances to prepare the nanocomposite electrode, which is applied as the anode or cathode to sodium-ion batteries. The recent research progress of graphene-based nanocomposites for SIBs is also summarized in this study based on the above categories, especially for nanocomposite fabricate methods, the structural characteristics of electrodes as well as the influence of graphene on the performance of the SIBs. In addition, the relevant mechanism is also within the scope of this discussion, such as synergistic effect of graphene with active substances, the insertion/deintercalation process of sodium ions in different kinds of nanocomposites, and electrochemical reaction mechanism in the energy storage. At the end of this study, a series of strategies are summarized to address the challenges of graphene-based nanocomposites and several critical research prospects of SIBs that provide insights for future investigations.

## 1. Introduction

Lithium-ion batteries (LIBs) are a kind of high-specific energy secondary battery that are important and widely used in commercial applications nowadays. However, due to the limitation of lithium resources, the cost of lithium-ion batteries has greatly increased recently, which will become a serious constraint for its future massive applications [1]. Therefore, sodium ion batteries (SIBs), have many advantages such as rich resources, low-cost, high-energy conversion efficiency, no over discharge characteristics, compatible with aluminum foil that make it have great application potential in new energy field [2,3]. In addition to partially replacing the highly polluting lead-acid batteries, the most significant potential application of sodium-ion batteries can also focus on balancing the peak and valley of electricity from the wind or solar power, as shown in Figure 1 [4]. People started the research on SIBs as early as the 1970s and it has witnessed a spurt of growth with the development of material science in the recent year [5]. The working principle of SIBs is similar to that of LIBs [6,7]. Its internal structure mainly contains four parts: Positive, negative, electrolyte, separator. Normally, both anode and cathode electrodes are composed of active material powders, adhesives and conductive additives in proportion and separated using a porous separator usually made of glass fibers or porous polymers that allows ions to pass through but prevents electrons that achieve short circuit prevention [6].

The working principle of sodium-ion battery is schematically shown in Figure 2. In essence, sodium-ion battery is an electrode with a concentration difference in ion concentration. In the process of charging the battery, the sodium ions are detached from the positive electrode, pass through the electrolyte, the separator, and transferred to the negative electrode material, the same time the electrons generated by the formation of sodium ions are passed from the outer circuit to the negative electrode, forming a closed circuit. During the discharge of the battery, the sodium ions are detached from the negative electrode and transferred to the positive through the electrolyte and separator, the electrons are also re-transferred from the negative electrode to the positive electrode of the battery through the external circuit, forming a closed loop road, to achieve charge conservation of positive and negative electrodes [8]. However, compared with lithium ion, sodium ion has a stronger solvation with the size of 1.02 Å (0.76 Å for the lithium ion) and the electrochemical potential of sodium-ion (−2.71 V, Na^+^/Na) is higher than that of lithium ion (−3.04 V, Li^+^/Li). Therefore, the electrode material of sodium-ion batteries will produce more remarkable volume changes in the charge and discharge process, resulting in more severe problems, including collapse and crushing of electrode structure. At the same time, from the perspective of dynamics, the insertion and extraction speed of larger sodium ions in electrode materials is slower than that of lithium ions, which makes sodium-ion batteries have more significant challenges in cycle life and rate capability performance [9,10].

Up to now, to solve the above limitations, an effective method has been proved for designing nanoscale structured materials, such as nanodots [11], nanosheets [12], nanotubes [13], nanorods [14,15], nanospheres [16], and nanocomposite structure composed of two or more of the above materials [17,18,19]. Considering the advantages of unique physicochemical properties, lots of exploration has triggered to realize the nanocomposites for functional energy storage applications. According to recent research, 0D nanosphere/nanoparticles, 1D nanowires/nanoneedles, 2D nanoplates/nanosheets and multi-level architectures of nanocomposite architectures have been established and shown to enhance the specific surface area, accessible pore channels, more expose active sites, and promoted ion and electron transfer [19,20,21,22]. Meanwhile, the nanocomposites have distinctive advantages such as facilitating the penetration of electrolytes, shortening the ion diffusion lengths and increasing the loading amount of active substances, resulting in improved rate ability, energy density and power density [19]. Li, M. et al., reported the nanocomposite of MnO_2_ ultrathin nanosheet fabricated on TiN nanowires modified carbon cloth to produce the core–shell binder-free flexible electrodes for ultra-fast supercapacitor and SIBs with high voltage window. The MnO_2_/TiN/CC electrode showed a discharge time of 120.2 s at a current density of 2 mA cm^−2^, while 35 s and 34.8 s for the electrode of TiN/CC and MnO_2_/CC, respectively. According to the above galvanostatic charge/discharge (GCD) results, MnO_2_/TiN/CC electrode obtained a specific capacity of 318.41 F g^−1^ at 2 mA cm^−2^, which is 2.7 times larger than that of MnO_2_/CC (120.5 F g^−1^). However, although the high porosity of the nanocomposites alleviated the volume expansion effect of materials caused by chemical reactions during the energy storage process to a certain extent, sodium ions have a larger ionic radius; this put forward a higher request for the structural stability of the composites [17,23]. Moreover, many active sodium storage materials, such as bimetallic oxides, have poor electrical conductivity [24]; organic compounds, have voltage platform and poor conductivity [25]; Prussian blue derivatives, have low coulomb efficiency and poor conductivity [26]; polyanionic compounds, have low reversible capacity ratio and some contain toxic elements [27]; which further affect the overall performance of SIBs. Therefore, nanocomposites with the advantages of avoiding the shortcomings of active materials are proposed to improve the overall electrochemical performance of SIBs. Nongkynrih J. et al., used sonochemically assisted sol gel followed by ball milling method constructed W-doped Na_3_V_2_(PO_4_)_2_F_3_@C as cathode material in sodium ion batteries [28]. Yi, T.F. et al., studied NiCo_2_S_4_-based nanocomposites for energy storage [29]. Sandwich architecture of graphene/MoS_2_composite were synthesized successfully by Yang, Z. et al., used a simple evaporation and hydrothermal reaction method as anodes for enhanced reversible lithium and sodium storage [30].

Due to the advantages such as low price, environmental friendliness, ion storability and high conductivity, carbon materials have become the optimal selection in the range of potential SIBs electrode materials [31]. Carbon materials manifest themselves as various hybridized forms of sp, sp^2^, and sp^3^ that produce many allotropes with different physicochemical properties [31,32,33,34]. Graphene, the typical representative of the sp^2^-hybridized carbon materials composed of six-membered rings with a honeycomb structure, shows a broad application in catalysis, sensor, energy storage and other fields [35,36,37,38]. Graphene and its derivatives are widely studied. Graphene can be prepared by mechanical stripping, vapor deposition, epitaxial growth, redox and other methods [39,40,41,42]. The main preparation methods of graphene oxide include Brodie method, Staudenmaier method and Hummers method, which are all prepared by the oxidation of graphite by strong oxidizing agent in concentrated acid medium [43]. The Brodie method prepared graphene by using fuming HNO_3_ and KClO_3_, the Staudenmaier method modified Brodie’s method by replacing about two-thirds of the fuming HNO_3_ with concentrated H_2_SO_4_, and the Hummers method used concentrated H_2_SO_4_, NaNO_3_ and KMnO_4_ to react with graphite to produce graphite oxide. The preparation methods of redox graphene mainly include chemical reduction, electrochemical reduction, thermal reduction, etc. [44,45,46]. Benefitting from the excellent electrical conductivity, high specific surface area, structure controllable and good compatibility with other materials, graphene and its derived structures, such as graphene aerogel, stacked graphene layers and graphene foam have been used as the electrode for SIBs [47]. However, the pristine graphene has certain limitations in energy storage and insertion/deintercalation of sodium ions process because it does not provide sufficient active sites and the ideal amount of functional groups for enough interaction with sodium ions [31,48,49].

To solve the above problems, compatible the above-introduced nanoscale active materials with graphene is a feasible way to simultaneously stimulate the storage of energy properties of both active and carbon materials, thus ensuring fast ion/electron transfer and utilization rate of the active substance with excellent conductivity. More importantly, graphene functions as physical barrier that further buffers volume expansion and contraction of nanomaterials during the sodiation/desodiation. In addition, the synergistic effect confinements between graphene and nanoscale active materials can efficiently suppress the aggregation of both the energy storage nanomaterials and graphene during the process of material synthesis and energy storage process [18]. For instance, Ma, J.Y. et al., constructed the nanocomposite of TiO_2_/MoS_2_ nanoflake linking with reduced graphene oxide (TiO_2_/MoS_2_/RGO). Compared with MoS_2_ (200 mA h g^−1^ at 5 A g^−1^) and TiO_2_/MoS_2_ (250 mA h g^−1^ at 5 A g^−1^), the nanocomposite showed a high specific capacity of 310 mA h g^−1^ at a current density of 2 A g^−1^ and ultrahigh structural integrity after 350 cycles [50]. The synergistic effect among RGO, MoS_2_ and TiO_2_ provides effective ion diffusion and electron transfer spaces and promotes insertion/deintercalation of Na ions that lead to the high performance of SIBs [50].

Therefore, to achieve the practical applications of sodium ion storage, various graphene-based nanocomposites fabricate stargates have been explored and can be mainly divided into the following categories according to the characteristics of the nanocomposites’ structure: graphene surface modification and doping [51,52], three-dimensional structured graphene [53], graphene coated on the surface of active materials [54], the intercalation stacked two-dimensional graphene [55,56], as shown in Figure 3. Through one or more of the above strategies, graphene is compounded with active substances to prepare the nanocomposite electrode, which is applied as the anode or cathode for SIBs. In this study, the progress of graphene-based nanocomposites for SIBs is reviewed, especially for nanocomposites fabricate methods, the structural characteristics of nanocomposites as well as the influence of graphene on the performance of the SIBs based on the above categories. In addition, the relevant mechanism is also within the scope of discussion, such as the Na^+^ storage process in different kinds of nanocomposites, the synergistic effect of graphene and active substances, electrochemical characteristics in the energy storage process. At the end of the overview, a series of strategies are summarized to address the challenges of SIBs and several possible research directions on graphene-based nanocomposites that provide insights for future investigations.

## 2. Literature Review According to the Structure of Graphene-Based Nanocomposites

### 2.1. Graphene Surface Modification and Doping

The preparation of materials and structural design is fundamental aspects that should be paid attention to enhance electrochemical performance for SIBs. The active particle size, electrode porosity, conductivity and electrolyte choices together primarily determined the rate capability, energy density and cycling ability of the electrode materials [24]. Therefore, graphene as the electrically conductive and morphologically controllable functional material, which is very helpful for facilitating rapid electron and ion transportation by constructing an excellent conductive 2D layer or 3D network throughout the electrode [61]. The introduction of atoms such as N, S, P and B as active dopant sites into the graphene as the functional groups improved the chemical and electrical characteristics of the electrode and promoted practical energy storage applications [62,63]. Furthermore, the electrochemical abilities of SIBs can be further enhanced by uniformly depositing and bonding the active substance of nanoparticles on the interface or layer of graphene that modified the chemical properties of electrode. Hence, based on combining the graphene treatment methods of doping and surface modification with active materials, the conductivity accommodates the volume expansion ability, and the electrochemical performances of the nanocomposite can be enhanced through the synergistic effects [64].

Chong, S.K. et al. deposited the ultrathin MoSe_2_ nanosheets perpendicularly onto the surface of reduced graphene oxide (MoSe_2_@rGO) hydrothermally as the anode for SIBs [65]. By regulating the quality radio of graphene as 1, 5, 10, and 20% in the nanocomposite, the effects of graphene on the performance of SIBs were compared in this study. Figure 4 shows the process of preparing MoSe_2_ and MoSe_2_@rGO composites by hydrothermal method with selenium powder as selenium source and sodium molybdate as molybdenum source [65]. This unique heterogeneous composite of MoSe_2_@rGO has a strong C−O−Mo and C−Mo interfacial bonding connection, so that the distance of crystal between MoSe_2_ layers is expanded to 7.9 Å, which facilitates stability of structure from considerable volume changes and obtained fast electron and alkaline ion transport capability.

The electrochemical performance of graphene based MoSe_2_ SIBs were assessed from 0.01 to 3.0 V. As shown in Figure 5a, MoSe_2_@5%rGO as a representative for cyclic voltammetry (CV) tests to investigate the electrochemical reversibility and redox process for SIBs. During the initial CV process, five prominent peaks can be observed, among which the peaks at 1.88, 1.35, and 0.80 V corresponding to the Na^+^ ions insertion into the layer of MoSe_2_ to produce Na_x_MoSe_2_, which is further transformed into Mo and Na_2_Se [66]. The peaks at 1.07 and 0.46 V correspond to solid state interphase (SEI) film formation and electrolyte decomposition [66]. The anodic peaks from the first to third cycle and the cathodic peaks from the second to third cycle remain at almost the same voltages, indicating excellent chemical and structural stability of the lamellar composites [66]. Figure 5b reflects the galvanostatic charge/discharge (GCD) curves of MoSe_2_@5% GO at 100 mA·g^−1^ in which the plateaus match well with the CV results and the GCD curves are well overlapping after the first GCD test that manifests the reversible electrochemical properties. The first discharge/charge specific capacity of MoSe_2_@5%rGO can be achieved to 712.7/458.3 mAh g^−1^ with the Coulombic efficiency of 64.3%, which is better than the specific capacity of MoSe_2_ (622.3/338.1 mAh g^−1^), MoSe_2_@1%rGO (798.9/495.8 mAh g^−1^) and MoSe_2_@10%rGO (716.9/355.2 mAh·g^−^^1^). It also indicates that the composite materials possess highly reversible specific capacities and a good cyclic stability. Figure 5c depicts that the MoSe_2_@5%rGO achieved the largest retention value of 383.6 mAh·g^−1^ after 50 cycles, which is better than the cycling stability of other rGO content. Compared with the other rGO mass content radio in Figure 5d, under the current density of 200, 500, 1000, 2000, and 5000 mA g^−1^, the MoSe_2_@5%rGO shows the best rate capability with the capacities of 447.4, 390.0, 348.5, 306.9, and 251.3 mAh g^−1^, respectively. In addition, after the above GCD rate capability tests, the MoSe_2_@5%rGO maintained an excellent reversible capacity of 435.3 mAh g^−1^ when the current density came back to 100 mA g^−1^. Therefore, MoSe_2_@5%rGO exhibits the optimal Na-ion storage performances by taking advantage of great kinetics and alleviating volume expansion effect from graphene and high capacity from MoSe_2_ [65].

Wang, S.H. et al., fabricated the Fe_2_O_3_ nanoparticles on the surface of S, N co-doped reduced graphene oxide (denoted as SNRGO) (Fe_2_O_3_/SNRGO) as the SIBs anode by microwave-assisted method [56]. The nanocomposites prepared through the microwave-assisted method have some advantages, such as low cost, Fe_2_O_3_ combined closely with graphene and high preparation efficiency, which makes it possible to produce the electrode on large scale [56]. Figure 6a,b clearly show that Fe_2_O_3_ nanoparticles with a diameter of around 30–80 nm is uniformly distributed on the interface of SNRGO that prevent the agglomeration of Fe_2_O_3_ nanoparticles and further reducing the volume change of Fe_2_O_3_ during the Na^+^ insertion/extraction [67]. Consistent with SEM images, the TEM images of Fe_2_O_3_/SNRGO nanocomposite in Figure 6c,d confirmed that Fe_2_O_3_ nanoparticles are deposited on the surface of SNRGO nanosheets tightly, which should be the main reason result in excellent cycle performance of SIBs.

The electrochemical behavior, including CV, GCD, rate capability performance and cycling stability of Fe_2_O_3_/SNRGO electrode was investigated through a half cell configuration. The CV curves of Fe_2_O_3_/SNRGO were carried out with a voltage window of 0.01–3 V under the scanning rate of 0.1 mV s^−1^ in the initial four cycles (Figure 7a). The two irreversible reduction peaks of 0.5 V and 1.1 V at the CV curves in the first negative scan of Fe_2_O_3_/SNRGO correspond to the irreversible formation of the SEI layer [68], meanwhile the reduction peak at about 0.01 V attributed to Fe^3+^→Fe^0^ [69]. In the following three CV scans, the two reduction peaks have moved to 0.63 V and 0.92 V, and in the oxidation direction, two peaks can be found at 0.66 V and 1.4 V that correspond to the reversible redox reaction of Fe^3+^↔Fe^2+^ and Fe^2+^↔ Fe^0^ [70,71]. Moreover, the CV curves are well overlapping after the first CV cycle that manifests the reversible electrochemical behavior of the Fe_2_O_3_/SNRGO electrode. The typical GCD curves of 1st, 10th, 20th, 30th, 40th and 50th of Fe_2_O_3_/SNRGO based half-cell can be found in Figure 7b. In the first GCD curve, the as-prepared sample manifests a large irreversible capacity with the Coulombic efficiency of only 29.0%, which can be explained as the formation of the SEI layer due to the irreversible insertion of Na^+^ into the lattice [72]. It is worth noting that no obvious discharge platform has been found in the GCD curve, which is consistent with the voltage trend of the other typical Fe_2_O_3_ based reports [67,72,73].

### 2.2. The Three-Dimensional Structured Graphene for SIBs

Recently, three-dimensional structured graphene as the current collector [57] compatible with the energy storage active materials [74] were used as SIBs electrode that demonstrates excellent electrochemical performance due to the synergistic effect and large surface area. The traditional process is common and uses two-dimensional materials such as copper foil and aluminum foil as the substrate, on which the active substances are coated for SIBs electrodes. Thus, in order to bind the active substances tightly on the two-dimensional foil surface, the active substance is first mixed with carbon material and binder to make slurry, and then coated on the foil [75]. Therefore, it is necessary to build a three-dimensional structure on which the active material can be firmly deposited by physical or chemical methods without the usage of binder to form the electrodes for rapid ions and electrons transportation. Due to the plasticity of graphene, the 3D structured architecture of graphene has been designed to reduce the restacking of graphene sheets during the electrode fabrication and electrochemical process and keep the large contact area with the electrolyte for enhancement of Na^+^ storage capacity with excellent cycling stability [76]. In addition, additive-free composite not only can enhance conductivity, simplify the preparation process, reduce the cost [77], but also can meet the requirements of large specific surface area for energy storage, higher capacity, and mechanical strength [78].

Therefore, an additive-free composite of the energy storage substances with a large specific surface of three-dimensional graphene current collectors for sodium-ion storage is an ideal method to realize the next generation of high-performance SIBs with higher capacity and mechanical intensity [57].

Figure 8a depicted the fabrication procedure of hierarchical periodic thickness adjustable microlattice with micrometer pores and submillimeter filaments/channels carried out by Yan, J. et al., through a simple 3D printing method using reduced graphene oxide/carbon nanotube (rGO/CNT) aerogel [47]. As shown in Figure 8b–d, the 3D rGO/CNT microlattice aerogel has a strong electrical conductivity and abundant active nucleation sites for Na plating/stripping with a reduced current density, resulting in uniform sodium deposition to overcome the issues of dendrite. Moreover, the ion transportation rate also greatly improved due to the ordered microchannels that existed in the artificial periodic microlattice that facilitate the electrolyte permeation into the microlattice and promote the contact between active substance and electrolyte. Intrinsically, the CNTs were stable scaffolds by the rGO that improved the surface kinetics and overall conductivity synergistically, further improve the distribution of energy storage substances uniformity as well as the overall performance of the composite electrodes.

To evaluate the practical performance of rGO/CNTs microlattice aerogel for SIBs, Na metal were electrodeposition on the surface of rGO/CNT as the anode (Na@rGO/CNT) and Na_3_V_2_(PO_4_)_3_ mixed with carbon on rGO (NVP@C-rGO) as the cathode were assembled into a full sodium ion battery (Figure 9a). Figure 9b depicted that under the current density of 100 mA g^−1^, the NVP@C-rGO cathode achieved a charge/discharge capacity of 98.8/91.1 mA h g^−1^. The performance of SIBs was also evaluated within the potential window of 2.0–3.8 V and the charge/discharge capacities of the full battery are calculated to be 101.3 mA h g^−1^/93.1 mA h g^−1^ at the 7th cycle (Figure 9c,d). After the active process of the first five cycles at 20 mA g^−1^, the cycling performance was carried out at 100 mA g^−^^1^ and delivered good cyclic stability during 100 cycles. The long-term cycling stability of the Na@rGO/CNT//NVP@C-rGO SIBs demonstrate that the 3D printed rGO/CNT microlattice has a great potential for practical application.

Wu, D.J. et al., reported the multi-layer nanostructured graphene modified 3D macroporous electrically conductive network (n-MECN) as the sodium ion battery substrate with superior electrochemical performance [24,79,80]. As described in the previous report and illustrated in Figure 10, the multilayer nanographene was prepared on the surface and inter-channel of MECN by solvothermal carburization method, from which the ordered porous-rich electrode substrate of n-MECN was realized [81]. In addition, then the self-branching MnCo_2_O_4_ was hydrothermally in situ growth on the surface and inter-channel of n-MECE at 180 °C to fabricate the MnCo_2_O_4_@n-MECN electrode. The special ordered structure with micro-channel of n-MECN not only acts as the skeleton in the electrochemical energy storage process and facilitate the transfer of electrolyte but is also compatible with the microelectronic process for the micro energy storage devices.

Figure 11a,b reveal that the nanographene layers on the surface of MECN have a large number of ripples and wrinkles with a lateral size of 50–200 nm to realize the 3D electronic conductivity substrate. The 3D interconnected structure of n-MECN inherited from MECN improved the electronic conductivity and the specific surface area to facilitate electrochemical reactions. The self-branching MnCo_2_O_4_ nano-architecture is incorporated into the surface (Figure 11c,d) and cross-sectional (Figure 11e,f) of n-MECN with a diameter around 50 nm, indicating that the MnCo_2_O_4_/n-MECN was successfully prepared.

To evaluate the practical application of the materials and structure design strategy, the MnCo_2_O_4_@n-MECN served as the anode in SIBs. In the cathodic process of the first CV curves cycle shown in Figure 12a, the two reduction peaks can be found at 0.25 V and 0.87 V, which are related to the irreversible formation of the SEI layer and the process of Mn^2+^ to Mn, Co^3+^ to Co^2+^, Co^2+^ to Co [82]. The two broad peaks related to metallic Mn to Mn^2+^ and Co to Co^2+^, Co^2+^ to Co^3+^ are anodic oxidation processes at 0.84 V and 1.59 V [83,84,85]. After the first CV cycles, the CV curves are well overlapping and the moves of the cathodic peak to 0.87 V related to the formation of the Na_x_MnCo_2_O_4_ phase, demonstrating reversible electrochemical behavior of MnCo_2_O_4_@n-MECN electrode [86]. Figure 12b shows the 1st, 2nd, 3rd, and 50th GCD curves of the sodium ion battery at 50 mA g^−1^. A noticeable discharging plateau at ~0.45 V can be found in the GCD curves corresponding to the reduction of MnCo_2_O_4_ nanoarrays which is in accordance with the morphologies of CV curves [87]. The charging/discharging specific capacity calculate from the 1st GCD curve of MnCo_2_O_4_@n-MECN reaches 614.7/1120.0 mAh g^−1^ with the Coulombic efficiency of 75.1% [88,89]. Figure 12c reflected that MnCo_2_O_4_@n-MECN based SIBs show the specific capacities of 541.2, 200.5, 150.1, 110.0, and 70.5 mAh g^−1^ at the current densities of 50, 100, 200, 400, and 800 mA g^−1^, respectively, and the reversible capacity is maintained when the current density back to 50 mA·g^−1^. As shown in Figure 12d, the SIBs reflected a good cycling ability of 100 cycles under different current densities after 1st cycle as well as the morphology and structure of MnCo_2_O_4_@n-MECN in SIBs are maintained after 100 cycles. Figure 12e depicted that the b value of MnCo_2_O_4_@n-MECN based SIBs is 0.81 at the cathodic peak indicating that the energy storage mechanism of the SIBs involves both faradic and non-faradic processes and the capacitive-controlled process makes the main contribution to the energy storage characteristics of MnCo_2_O_4_@n-MECN. With the increase of the scan rates the contribution of capacitive-controlled gradually increased, when scan rates from 0.2 mv s^−1^ to 1.0 mv s^−1^, the capacitive contribution improved from 52% to 73%, and when the scanning speed is greater than 0.8 mv s^−1^, the capacitance contribution rate hovers around 70% (Figure 12f). This study fully demonstrates that the three-dimensional structured graphene exhibits excellent electrochemical properties, optimizing ion and electron transport, and improves the utilization of active material so as to increase the rate capability ability of the SIBs [85,86,87].

Jo, M.S. et al. described an electrode of graphene-based nanofibers entangled with bamboo-shaped N-CNTs containing CoSe_2_ nanocrystals at each node (N-CNT/rGO/CoSe_2_ NF) through the heat treatment process of electrospinning nanofibers [90]. Figure 13 describes the detailed formation mechanism and fabrication steps of this nanostructured electrode. An electrospinning process (Figure 13(a①)) was used to fabricate the precursor nanofibers that were composed of graphene oxide (GO), Co salt, poly (vinyl alcohol) (PVA), and PS nanobeads (40 nm). Next, the highly integrated structure of mesopores rGO nanosheets mixed with carbon-covered Co nanometals structure (Figure 13(a②)) were formed through the initial step of the first heat treatment of the precursor nanofibers with DCDA at 400 °C under N_2_ atmosphere. This step initially produced Co nanometals surrounding with carbon formed by PVA decomposition that prevents the aggregation of Co nanometals. Meanwhile, during the initial step of the first heat treatment step, the uniform mesopores between the GO sheets were formed due to the decomposition of PS nanobeads distributed in the GO matrix and the GO matrix was reduced to rGO at the same time. The uniform-sized Co nanoparticles were uniformly formed in the structured mesopores when the reducing gas penetrates into the depth of the structure from the mesopores.

As shown in Figure 13(a②), the further step of the first heat treatment was carried out at 800 °C under the CH_x_ and NH_3_ gases atmosphere and the bamboo-like N-CNTs were grown in the inner and outer parts of the rGO matrixed nanofibers by using the metallic Co nanocrystals in the mesopores as catalysts via catalytic chemical vapor deposition [91]. Interestingly, N-CNTs formed by aggregation of the C and N atoms from CH_x_ and NH_3_ gases that adsorbed and dissolved on the surface of the Co nanoparticles and the Co nanoparticles move forward by the force arising from the growth of CNTs, so that Co nanoparticles were uniformly distributed at the nodes of bamboo-like N-CNTs. Since H_2_Se gas cannot penetrate the C layer of CNTs, it is difficult to convert the CNTs coated Co nanoparticles into CoSe_2_ directly by high temperature gas phase method, as shown in Figure 13b. Therefore, part of the CNT walls surrounding the Co nanoparticles were thermally etched during the second heat treatment step at 300 °C for 30 min in an air atmosphere (Figure 13(a③)), although Co metal partially oxidized into Co_3_O_4_ during this step, that was ignorable and inconsequential. Finally, in the third heat treatment step, Co and Co_3_O_4_ nanoparticles inside the N-CNTs were selenization at 350 °C under H_2_Se gas atmosphere to form the CoSe_2_ nanoparticles (Figure 13(a④)). Via the above steps, the electrode of graphene-based nanofibers entangled with bamboo-shaped N-CNTs containing CoSe_2_ nanocrystals at each node (N-CNT/rGO/CoSe_2_ NF) was successfully obtained. As shown in Figure 13b, selenization of Co nanocrystals captured in the walls of carbon nanotubes is very difficult because H_2_Se gas used for selenization cannot penetrate the thick carbon nanotube walls. Therefore, the gas diffusion process and CO phase change can be achieved by introducing the defect part into the CNT wall during the intermediate heat treatment to achieve the goal of successfully generating CoSe_2_.

As shown in Figure 14a–c, the FE-SEM image of N-CNT/rGO/CoSe_2_ NF was well-maintained throughout, even after multi-step fabrication. From the HR-TEM image of Figure 14d,e, the authors found that the diameters of the nodes in the bamboo-like N-CNTs were around 5–10 nm. The lattice spacings of 0.22 nm match well with the (111) crystal planes of CoSe_2_ in N-CNTs (Figure 14e) [92] and the specific peaks of the XRD pattern of N-CNT/rGO/CoSe_2_ NF match well with the PDF#53-0449 of CoSe_2_ (Figure 14f) that indicating the complete selenization of Co and Co_3_O_4_ into CoSe_2_ [92,93].

In Figure 15, the electrochemical performance of the two electrodes of N-CNT/rGO/CoSe_2_ NF and CoSe_2_-filled NF as the anode for SIBs were compared using carbonate-based electrolytes. At a scanning rate of 0.1 mV s^−^^1^, the initial five CV cycles of the two SIBs were carried out under the potential window of 0.001–3.0 V (Figure 14a,b). For the above two samples in the first cycle, the peak of ≈0.78 V in discharging process owing to the formation of metallic Co nanocrystals and Na_2_Se from the conversion reaction between the Na^+^ and CoSe_2_ as well as the anodic peaks at 1.84 and 1.96 V owing to the restoration of metallic Co and Na_2_Se into CoSe_2_ nanocrystals in the charging process [90,92,94]. Moreover, after the first CV cycle, the insertion of Na^+^ into CoSe_2_ corresponded to the cathodic peaks at 1.41 and 1.11 V, and due to the formation of ultrafine CoSe_2_ nanocrystals, the peak shift to higher potential from the second cycle [90,92,94]. Moreover, the conversion reaction with further Na^+^ insertion was manifest at the cathodic peak of 0.7 V in the consequence CV cycles [90,94]. Figure 15c,d show the GCD curves of the N-CNT/rGO/CoSe_2_ NF and bare CoSe_2_-filled NF based SIBs at a current density of 0.5 A g^−1^ for the 1st and 2nd cycles. Typically, anode materials with crystalline materials exhibit flat plateaus and ultrafine nanocrystals or amorphous phases exhibit sloped charge/discharge profiles [95,96]. In the first discharge curve, the N-CNT/rGO/CoSe_2_ NF manifested a comparatively short plateau at ≈0.89 V with a sloped charge/discharge profile due to the ultrafine crystallite size of the CoSe_2_. While the bare CoSe_2_-filled NF showed a lower and flatter plateau compared with CNT/rGO/CoSe_2_ NF composite electrode due to the crystalline materials of CoSe_2_ in CoSe_2_-filled NF. Bare CoSe_2_-filled NF shows the increase of resistance in the electrolyte due to the polarization from slow Na^+^ ion diffusion, while the nanostructure of N-CNT/rGO/CoSe_2_ NF has an increased Na^+^ ion diffusion rate results a more easily Na^+^ insertion/desertion reactions [97,98]. The initial discharge capacities of the N-CNT/rGO/CoSe_2_ NF was 645 mA h g^−1^ with Coulombic efficiencies of 72.4%, while the bare CoSe_2_-filled NF was 531 mA h g^−1^ with the Coulombic efficiencies of 83.5%, although the active materials in N-CNT/rGO/CoSe_2_ NF were smaller than the bare one.

In order to investigate the long performance of the SIBs based on N-CNT/rGO/CoSe_2_ NF and bare CoSe_2_-filled NF, the cycling performance was carried out at a current density of 0.5 A g^−1^ (Figure 15e). The bare CoSe_2_ based SIBs exhibited a significate capacity fading that only 34 mA h g^−1^ after the 100th cycle because of volume change and structural destruction of CoSe_2_ during repeated insertion/extractions of Na^+^ ion. In contrast, the N-CNT/rGO/CoSe_2_ NF based SIBs showed the discharge capacities of 512 and 400 mA h g^−1^ at 2nd and 100th cycles, indicating the constant cycling ability up to 100th cycles. Due to the involvement of graphene, the vertical interconnection between graphene sheets and N-CNTs embedded with CoSe_2_ nanoparticles both restrained the aggregation and buffered the volume expansion of CoSe_2_ nanoparticles during the cycling. To investigate the rate capacity of the nanocomposite, at current densities of 0.5, 1, 2, 5, 10, and 20 A g^−1^, the N-CNT/rGO/CoSe_2_ NF achieved the discharge capacities of 516, 470, 420, 341, 318, and 215 mA h g^−1^, respectively (Figure 15f). Furthermore, the discharge capacity was maintained at 512 mA h g^−1^ when the current density back to 0.5 A g^−1^ after 60 cycles. In contrast, as the increased of current density from 0.5 to 20.0 A g^−1^, the discharge capacity of the bare CoSe_2_-filled NF decreased from 388 mA h g^−1^ to almost zero.

### 2.3. Graphene Coated on the Surface of Active Materials

In recent years, nanomaterials uniformly encapsulated by graphene as the Na-ion battery electrodes with shortened diffusion lengths of electrons/sodium ions have been widely studied and the graphene network formed the shell enables a highly conductive matrix. The graphene shell also provides enough physical space for the volume expansion of coated nanoparticles and at the same time constrains the nanoparticles within the conductive graphene shell for stable working conditions [99]. As expected, the fabricated nanocomposites manifest outstanding electrochemical performance with both remarkable rate ability and long cycling ability. Additionally, the carbonaceous conductive graphene materials provide the pathways for electron transport that facilitate the reaction kinetics, increasing the Coulombic efficiency and stabilizing the electrode-electrolyte interfaces [58,100,101].

Tan, X.Y. et al., reported a nanocomposite based on Sn_4_P_3_ nanoparticles coated with porous graphene tubes (GT) as the Sn_4_P_3_/GT for the anode of SIBs [58]. The graphene tube is composed of two layers of graphene, the inner layer is nitrogen doped hydrophilic graphene tube and the outer layer is un-doped hydrophobic graphene tubes that provide the accommodation for the volume expansion of the coated Sn_4_P_3_ nanoparticles. N-doping not only increases the hydrophilicity contributes to the directional growth of nanomaterials and improves the electron transfer kinetic performance, but also the synergistic effect of N and graphene keeps the electrode structure stable and further improves the electrochemical performance. As illustrate in Figure 16, the graphene tubes are firstly dispersed in the aqueous solution containing tin precursor and the aqueous solution can penetrate into the hydrophilic layer of graphene tube easily. After that, the SnO_2_ nanoparticles prepared in the inside of the graphene tubes by selective growth of hydrothermal method and then a phosphidation process was used to forming the Sn_4_P_3_/GT.

Figure 17a depicted that in the voltage window of 0.01–2 V (versus Na/Na^+^) at 0.5 A g^−1^, the electrode of Sn_4_P_3_/GT shows a first GCD curve of charge/discharge capacities of 1002 mA h g^−1^/722 mA h g^−1^ with a Coulombic efficiency of 72.2%. The formation of SEI film is the main reason for the irreversible capacity during the first GCD cycle [102,103]. To realize the real application of the Sn_4_P_3_ based nanocomposite electrode, optimization of electrolyte and further exploring SEI composition should be carried out to reduce the initial irreversible loss for a higher Coulombic efficiency beyond 90% [104]. A capacity of 713 mA h g^−1^ has been realized after 500 cycles with well overlapping GCD curves, indicating the reversible electrochemical behavior of the Sn_4_P_3_/GT electrode.

Figure 17c depicted that the Sn_4_P_3_/GT electrode showed a good rate performance with the discharge capacities of 821, 722, 636, 547, 435, 386 and 326 mA g h^−1^ at current densities of 0.2, 0.5, 1, 2, 5, 10 and 20 A g^−1^, respectively. Additionally, the discharge capacity is maintained of 808 mA h g^−1^ when the current density is back to 0.5 A g^−1^ that indicating the recoverability of the electrode. In contrast, the discharge capacity decreased from ~650 mA h g^−1^ to almost zero of the bare Sn_4_P_3_ as the current density increased from 0.2 to 20.0 A g^−1^. The electrochemical impedance spectroscopy (EIS) was employed to further explore the performance of Sn_4_P_3_/GT and bare Sn_4_P_3_ electrode with the frequency from 0.01 Hz to 10 MHz and a perturbation amplitude of 10 mV. From the Nyquist plot of Figure 17d, the Sn_4_P_3_/GT electrode shows a much smaller diameter of semicircle compare with bare Sn_4_P_3_ electrode which indicate that the Sn_4_P_3_/GT composite electrode has a lower interfacial resistance as well as better kinetics and higher ion-diffusion rate during the process of sodiation/desodiation [105,106]. The active material utilization rate versus charge-discharge current density of the Sn_4_P_3_/GT and bare Sn_4_P_3_ electrode were investigated. As shown in Figure 17e, the utilization rate of the active material in Sn_4_P_3_/GT decreases with the current density, which is 88%, 77%, and 67% at 0.5 A g^−1^, 1 A g^−1^ and 2 A g^−1^, respectively. In contrast, the bare Sn_4_P_3_ electrode shows the active material utilization rate of 79%, 28%, and 3% at 0.5 A g^−1^, 1 A g^−1^ and 2 A g^−1^, respectively. All of the above electrochemical properties indicate that the overall performance of Sn_4_P_3_/GT was significantly improved by the introduction of graphene tubes on the surface of Sn_4_P_3_ nanoparticles.

As shown in the outline in Figure 18a, the graphene-coated Na_2_._4_Fe_1_._8_(SO_4_)_3_ (denoted as NFS) nanograins (NFS@rGO microspheres) was successfully fabricated by Fang Y.J. et al., the 3D graphene microsphere network via a spray-drying route [99]. From the SEM images of Figure 18b–d, the Na_2_._4_Fe_1_._8_ (SO_4_)_3_ nanoparticles are homogeneously distributed between layers of 3D spherical graphene and the porous in 3D graphene network are well preserved. The overall electrochemical performance of the nanocomposite of NFS@rGO was improved because of the synergistic effect between surface coated 3D graphene and Na_2_._4_Fe_1_._8_(SO_4_)_3_ nanograins [99].

To evaluate the sodium-storage behaviors of the materials and structure design strategy, the NFS@rGO microspheres were investigated by assembling 2016-type coin cells. As shown in Figure 19a, during the initial CV cycle, two pairs of distinct peaks can be found, while in the subsequent cycles, three pairs of peaks appeared, and the above phenomenon matches well with the previous study [107,108,109]. Figure 19b reflects the 1st to 4th cycles of the GCD curves of NFS@rGO microspheres at 0.05 C and the 1st charge curve shows a different shape compared to other curves, which agrees well with the CV in Figure 19a and previous research [107,108]. The reversible specific capacity of NFS@rGO electrode reaches 100 mAh g^−1^ with the average working potential of 3.75 V, obtaining an energy density of 375 Wh kg^−1^ for the corresponding full cell. Similar to CV curves, the GCD curves are almost overlapped after the initial cycle that manifests the electrode’s reversible electrochemical behavior in subsequent cycles. Furthermore, it may be due to the formation of the cathode electrolyte interface layer and the irreversible decomposition of electrolyte, the Coulombic efficiency of the NFS@rGO has a certain reduction in the subsequent few cycles compared with the initial Coulombic efficiency of 89% [110].

Figure 19c depicted that the SIBs full cell demonstrates a superior rate performance compared with the previous works that the reversible discharge capacities of 100, 90, 84, 80, 76, 72, 67, 64, 60, 54 and 45 mAh g^−1^ at current rate of 0.05, 0.1, 0.2, 0.5, 1, 2, 5, 10, 20, 30 and 50 C, respectively [107,111,112,113]. Additionally, the discharge plateaus of the NFS@rGO dropped with the increase of current rate, while the working potentials are still maintained above 3 V at a current rate even up to 30 C, better than most of the relative Fe-based cathodes, such as FePO_4_ [114], NaFePO_4_ [115], NaFePO_4_F [116], and Na_2_FeP_2_O_7_ [117].

### 2.4. Intercalation Layer Stacked Graphene

As a promising carbon material, graphene has a series of advantages, such as high conductivity, large surface area and exceptional mechanical strength [118]. Therefore, a kind of intercalation layer stacked graphene embedded in active materials between graphene layer was designed to achieve a shortened electron/Na^+^ transport distance with improved reaction kinetics [59]. The graphene layer act as the conducting pathway for electrons and mechanical support for nanomaterials that are confirmed to have a strong affinity to active materials, thereby mitigating the fragmentation of substances during the reaction [119]. The graphene layer and the active materials formed a sandwich electrode have a mechanically robust “nest” between the graphene layer to induce uniform deposition of active materials, therefore, inducing a strong surface capacitive contribution with high-rate capability [119]. Additionally, in the traditional electrode preparing process, polymer binders of insulation were used to adhesion the active nanomaterials on the surface metal Cu or Al current collector, which limited the rate capability, flexibility and energy density of the active materials [3,120]. Furthermore, the interfacial bonding of the traditional electrode between active materials and metal current collectors have many limitations during the repeated sodiation/desodiation process that will cause damage to the electrode under the long-term volume expansion [118,121,122]. Therefore, constructing binder-free and freestanding electrode with layer stacked graphene sheets will obtain the mechanical flexibility, high lightweight and electrical conductivity electrodes that will effectively improve the overall performance of SIBs.

Liu, M.K. et al., designed and developed a 3D architecture by CNFs directly penetrating into graphene sheets (CNFIG) as the electrically conductive template and then the MoS_2_ nanoflakes were in situ deposited on the surface of the CNFIG aerogel as the MoS_2_@CNFIG hybrid for the anode of SIBs (Figure 20a) [118]. A versatile interfacial method was first carried out to depose the MoS_2_ on the surface of CNFIG and then high-temperature treatment of the product of the first step to obtain the MoS_2_@CNFIG anode. As shown in Figure 20d–f, the layered MoS_2_ nanoflakes were uniformly grown on both graphene layers and the CNF bridges that increasing the loading amount of the active substance. The vertically aligned channels in the inter of MoS_2_@CNFIG nanocomposite accelerate the penetration of electrolyte and facilitate the rapid transfer of sodium ions (Figure 20b). Since CNFIG is composed of carbon materials that provide an efficient path for rapid electron transport, facilitates the insertion/extraction ability of sodium ions.

The SEM image and corresponding EDX elemental mappings (C, Na, Mo, and S) of MoS_2_@CNFIG anode after 1000 cycles were applied to investigate the structural and element integrity of the MoS_2_@CNFIG anode. From the SEM image of Figure 21a, the porous of the MoS_2_@CNFIG were preserved, so that the ionic diffusion path and the volume expansion space for MoS_2_ nanoflakes can be guaranteed during the long-time cycling. All of the above properties undoubtedly contribute to the quick sodiation/desodiation ability and rapid transfer of sodium ions. The EDX elemental mappings of C, Na, Mo, and S elements can be detected in MoS_2_@CNFIG anode after 1000 cycles proving that the composition of the anode was well maintained, and some sodium ions were adsorbed by the anode during the chemical redox reactions. Figure 21c shows three assembled full cells based on MoS_2_@CNFIG anode in parallel powering the 12 LEDs which demonstrate the potential applications of the MoS_2_@CNFIG nanocomposite.

To evaluate the rate ability and long-term cycling ability (Figure 21d), the MoS_2_@CNFIG anode was cycled firstly under the current density of 1 A g^−1^ for 300 cycles and then 10 A g^−1^ for 400 cycles with high Coulombic efficiencies approaching ~100%. The MoS_2_@CNFIG anode provides a capacity of 322 mA h g^−1^ after 300 cycles at 10 A g^−1^ and 303 mAh g^−1^ in the 700th cycle. Additionally, the specific capacity was maintained at 421 mA h g^−1^ when the current density back to 1 A g^−1^ and can continue the cycle of 300 times, which indicates the recovery ability of the electrode. The excellent rate ability and cycling ability of MoS_2_@CNFIG nanocomposite may be ascribed to its structural stability, excellent conductive of CNFIG and the high utilization efficiency of MoS_2_ nanoflakes with large specific surface area. The Nyquist plots of the full cell were investigated after being cycled 2 and 1000 cycles and presented in Figure 21e. The EIS spectra of the MoS_2_@CNFIG anode were obtained after being cycled two and 1000 times. The R_ct_ value reflects the diameter of the semicircle in Figure 21e represents the charge-transfer resistance. The MoS_2_@CNFIG hybrid at 2nd cycle shows a smaller R_ct_ value of 100 Ω compared with the R_ct_ value of 162.3 Ω at the 1000th, indicating that the initial electrode has a lower interfacial resistance as well as better kinetics and higher ion-diffusion rate during the sodiation/desodiation process. Furthermore, the slope of the low-frequency region does not decrease much with the increase of GCD cycles, implying that the electrochemical performance of the electrode has not change significantly. From the TEM images in Figure 21f, the MoS_2_@CNFIG also shows clear lattice fringes with flake-like MoS_2_ structures (inset of Figure 21f) after long-time cycles. Finally, the author attributes the excellent electrochemical storage ability of the SIBs to the 3D nested structures of MoS_2_@CNFIG producing nanoreservoirs between adjacent MoS_2_ nanoflakes [121], the vertically aligned channels in CNFIG [122] and the MoS_2_ anchored tightly on the surface of CNFIG with good electrical contact (Figure 21g).

Yang, Z.X. et al., reported the hierarchically structured graphene/MoS_2_ composites hybrid sandwich structure (3DG-MoS_2_) synthesized by solution evaporation followed the hydrothermal reaction method [30]. At the synthesis stage of the sample, NH_2_CSNH_2_ to the system, under hydrothermal conditions, releases H_2_S gas through the reaction of NH_2_CSNH_2_ + 2H_2_O → 2NH_3_ + H_2_S +CO_2_. After the reaction with H_2_S, the molybdate in G/MCu nanocomposite was converted into MoS_2_ nanosheets [123]. In order to improve the sodium storage ability, the 3DG-MoS_2_ nanocomposite was fabricated by embedding the Cu_2_O nanoparticles on the surface of graphene sheets (G/MCu) and then assembling MoS_2_ nanosheets further on the surface of graphene sheets by hydrothermal reaction (Figure 22a) [30,124,125]. Figure 22b indicated the FE-SEM image of as-prepared G/MCu, from which we can find that the Cu_2_O nanoparticles have been embedded into the graphene layer with the formation of the rough surface. The cross-section FE-SEM images of G/MCu further indicated that Cu_2_O as the tiny particles distributed uniformly among the layers of graphene. After the hydrothermal reaction with thiourea, the molybdate in G/MCu nanocomposite was converted into MoS_2_ nanosheets that formed the sandwich layer structure of 3DG-MoS_2_, as shown in Figure 22d,e. The particular architecture of layer structured graphene accelerated the penetration of electrolyte (NaClO_4_ (1 mol/L) in DEC: EC (1:1 *v*/*v*) solution) and improved the specific surface of active material that facilitates the overall performance of SIBs.

As shown in Figure 23, the author further investigated the electrochemical performance of the 3DG-MoS_2_ electrode under different conditions. Na storage ability of the 3DG-MoS_2_ electrode was initially evaluated through CV curves of the 3DG-MoS_2_ electrode at the voltage range of 0.0 to 3.0 V versus Na/Na^+^ under the scan rate of 0.5 mVs^−1^. As shown in Figure 23A, a wide peak of 0.8 V can be observed in the first cycle that may be related to the insertion of sodium ions into MoS_2_ and the irreversible formation of SEI layer [126,127]. Furthermore, the CV curves are well overlapping after the initial cycle, demonstrating excellent reversible electrochemical behavior of 3DG-MoS_2_ electrode. Figure 23B reflects the 1st, 2nd, 5th, 10th, 50th and 100th cycles of the GCD curves of 3DG-MoS_2_ electrode at a current density of 0.5 A g^−1^ and the 1st charge curve shows a different shape compare with other curves, which match well with the CV in Figure 23A. The different shapes of the first GCD curve may represent the capacity loss due to the irreversible formation of the SEI layer.

To evaluate the long-term cycling ability (Figure 23C), the 3DG-MoS_2_ electrode provides a specific capacity of 524 mAh g^−1^ under the current density of 0.5 A g^−1^ after 100 cycles (the slight increase of capacity with cycling may be due to activation process), which higher than that of the G/MoS_2_ and MoS_2_ electrodes of 320 and 284 mAh g^−1^, respectively. In the author’s opinion, the improvement of the cycling ability of the 3DG-MoS_2_ electrode is mainly ascribed to the structure of the strong G/MoS_2_ thin layers existing in 3DG/MoS_2_ electrode, which prevents the damage caused by the volume expansion of active material. In order to further investigate the overall performance of 3DG-MoS_2_ electrode, further 300 long-term cycling ability and Coulomb efficiency evaluations of the 3DG-MoS_2_ electrode were carried out at a current density of 1 A g^−1^ in Figure 23D. Obviously, the 3DG-MoS_2_ electrode still provides a capacity up to 479 mA h g^−1^ even after 300 cycles and exhibits the coulombic efficiency of ~100% in the whole 300 cycles except for the first one.

## 3. Discussion and Prospects

Although the graphene-based nanocomposites have been well developed in the field of new energy, there are still several key problems that should be solved to achieve high performance sodium-ion batteries for practical applications. Firstly, although a relatively comprehensive structural system of SIBs has been developed based on graphene, it is necessary to carry out the research based on more positive materials with high energy density and power density, as well as negative materials with small volume change in the process of circulation [128]. Furthermore, further improving the cycle stability of the batteries are not only an important strategy to improve the performance of sodium ion batteries, but also the critical point to realize the large-scale application of sodium ion batteries for commercial application [129]. In addition, at present, the cost of graphene-based electrodes is relatively high, and the synthesis methods of composite electrode materials are relatively complicated. More efficient strategies for combining graphene with active substances should be developed and the theoretical model of SIBs materials should be established in accordance with the development of graphene-based composite electrodes [130]. With the progress of the above technologies, sodium ion batteries can first replace the lead-acid batteries with high pollution and low energy density, and then play a greater role in more aspects. From a macro and pragmatic perspective, more research efforts of SIBs should be focused on the following four points as the prospects of this review:(1)In-depth knowledge of Na+ storage mechanism

At present, the mechanism of sodium storage is still inconclusive and needs further study, especially on low pressure platforms, ultrafine pore and closed pore filling [131]. Further studies using in/ex situ techniques [132], as well as theoretical calculations and simulations [133], are needed to better understand the role of pore size distribution, openings and closures, surface functional groups, structural defects, and pseudo graphitic domains in the sodium/desulphurization process. Developing a deeper understanding of interaction mechanism between graphene and active materials is helpful to design more novel graphene-based nanomaterials with special structures and high capacity for SIBs [134].

(2)Reasonable structure and process design of electrodes

In order to further improve the performance of the anodes, methods such as physical and chemical activation can be used. Such as: graded pore structure, microwave activation [135], H_2_ reduction [136] and high current pretreatment [137], heteroatom doping [138], metallic NPs (or allowable materials) synthesis of nano-composite materials [139]. Some efforts should be exerted on investigating the effects of morphology, porosity, size, and defects of graphene-based electrodes to achieve better electrochemical performance [134]. Of course, in addition to some anode materials introduced in the article, it is also important to explore more efficient cathode materials and composite method of graphene based electrodes, such as NaxMO_2_ (M = Co, Mn, Ni) [140,141,142], V_2_O_5_ [143], V_2_O_5_·nH_2_O [144], MoO_3_ [145], Na_3_V_2_(PO_4_)_3_ (NVP [146,147], Na_2_MP_2_O_7_ (M = Fe, Co, Mn) [148,149,150] and NaFe(SO_4_)_2_ [151].

(3)Environmental, economic, and scaling-up aspects

By implementing low-energy processes and avoiding the use of expensive and/or hazardous chemicals, in other words, at relatively mild temperatures, SIBs can be promoted closer to commercialization by using cheaper and sustainable sources (even waste streams) such as heteroatom doping [152]. The achievement of high-quality, mass-produced, and low-cost graphene through facile methods is hard but urgent. In addition, the modification of the graphene with higher utilization efficiency should not be ignored [131]. The initial coulombic efficiency (ICE) of SIBs for both anode and cathode materials, is a key parameter for high performance SIBs, and the point is to increase the transport rate of the Na^+^ ions. Therefore, developing SIBs with high ICE and rate performance becomes vital to boost the commercialization of SIBs [153]. Improving initial coulombic efficiency (ICE) is due to some decomposition of electrolyte and also to the capture of irreversible sodium in pore, functional group and interlayer space. Optimizing the formulation of electrolyte and binder [141], HC engineering defects, surface area and functional groups [154], and a thorough understanding of the formation and function of SEI are essential to promoting a rational and effective electrode design [155].

(4)Full cell fabrication and commercialization of SIBs

Full-SIB devices assembled from graphene-based anodes or cathodes show promising prospects in practical applications and related performance [156,157,158,159,160,161,162,163,164,165,166]. Full batteries with graphene-based anodes have lower average discharge voltage when they have large capacity [156,157,158,159,160]. In contrast, a full battery consisting of graphene-based cathodes yields higher voltage, but its capacity and cycle capacity are unsatisfactory [165,166]. In future studies, it is worth considering the balance between voltage and capacity of all SIB based on graphene. Anodes and cathodes with high capacity, stability and suitable voltage can all achieve high performance SIBs. Further research on the application of excellent electrochemical properties in semi-batteries to full batteries and hybrid capacitors is very popular. In the case where the cathode does not provide unlimited sodium ions, it is necessary to study the anode in the full battery to evaluate its actual performance. In addition, electrolytes and additives are particularly related to the compatibility of anode and cathode materials. Building flexible SIBs is a trend of graphene-based materials [131].

## 4. Conclusions

This review systematically introduces the combination of graphene with active substances as the nanocomposite electrodes for sodium ion electrochemical energy storage. To achieve this target, four categories of graphene-based nanocomposites fabricate stargates have been introduced in this article. Based on those categories and the research results of recent years, this review focus on the fabricate methods and structural characteristics of the graphene-based nanocomposites and the influence of graphene and morphology on the performance of the SIBs. With the development of nanotechnology as well as the excellent conductivity and plasticity of graphene, nanocomposite fabricates methods needed to be extended and expanded to stimulate the development and application of electrochemical energy storage in a benign direction. The discussion on research frontiers and applications in the field of graphene-based SIBs provide evidence that the application of graphene in advanced electrode materials is still a research hotspot, and more ingenious structure design and energy storage mechanism analysis will be the future development on the direction of nanocomposites for SIBs. Although graphene-based materials have been widely studied in SIBs and show good development prospects, their practical applications still face enormous challenges. In the first place, high-quality, large-scale production and low-cost graphene by simple methods are the basis for the production of graphene-based sodium ion batteries and should be further studied. In another, modification of graphene with higher utilization efficiency and better understanding of the mechanism of interaction between graphene and active materials is the basis of designing new electrodes. Studying the relationship of the performance of SIBs between the morphology, porosity, size and defect of graphene-based composite electrode is the key to improve the cycle life, high Coulomb efficiency and high energy efficiency of SIBs, which is the key to the marketization of SIBs. Furthermore, reaction mechanisms and theoretical analysis associated with Na^+^ insertion/depletion need further exploration.

## Figures and Tables

**Figure 1 nanomaterials-12-02837-f001:**
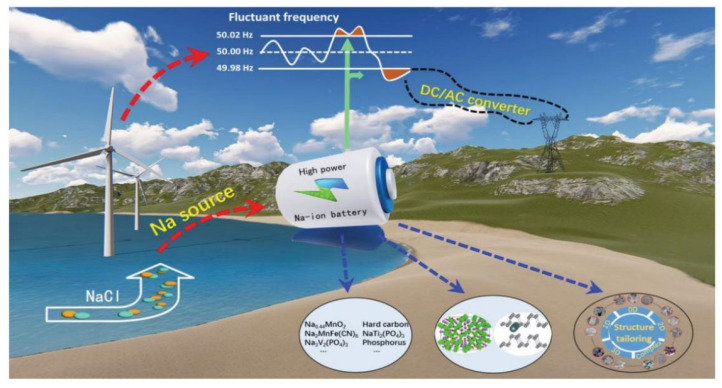
The blueprint of SIBs used in the high-power energy storage system. Reproduced with permission [4]. Copyright 2019, Wiley-VCH.

**Figure 2 nanomaterials-12-02837-f002:**
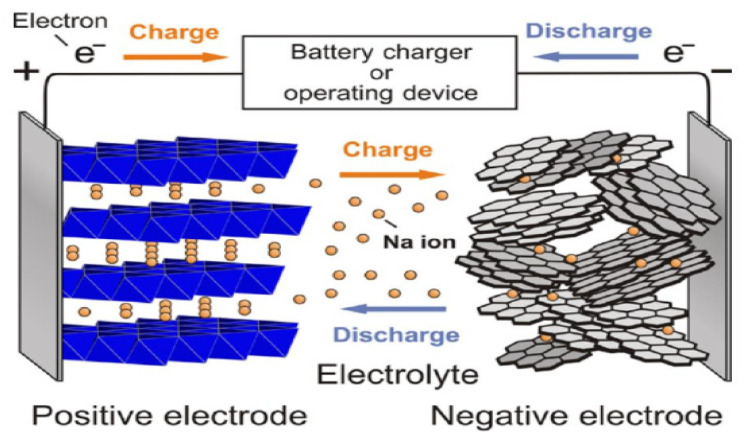
Schematic diagram of the working principle for sodium ion battery [8]. Copyright 2011, Wiley-VCH.

**Figure 3 nanomaterials-12-02837-f003:**
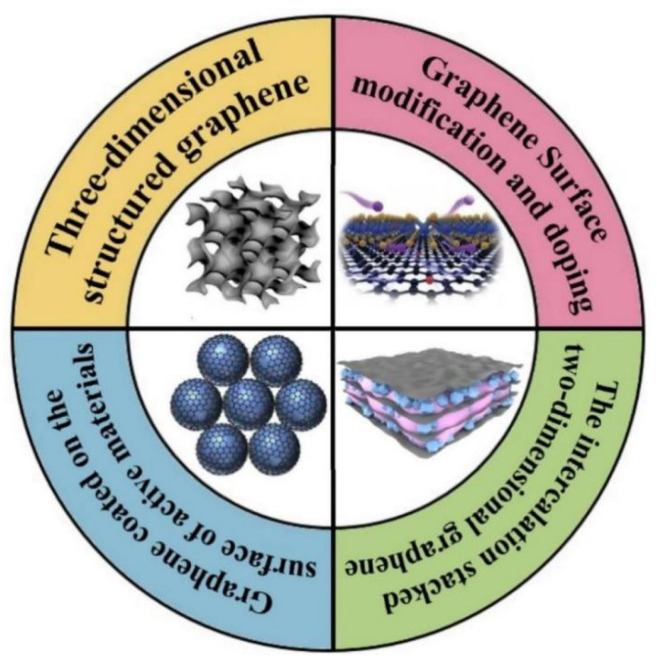
The main categories of graphene-based nanocomposites discussed in this study. Reproduced with permission [57,58,59,60]. Copyright 2018, Copyright Elsevier; 2021, Wiley-VCH; Copyright 2020, Elsevier; Copyright 2022, American Chemical Society.

**Figure 4 nanomaterials-12-02837-f004:**
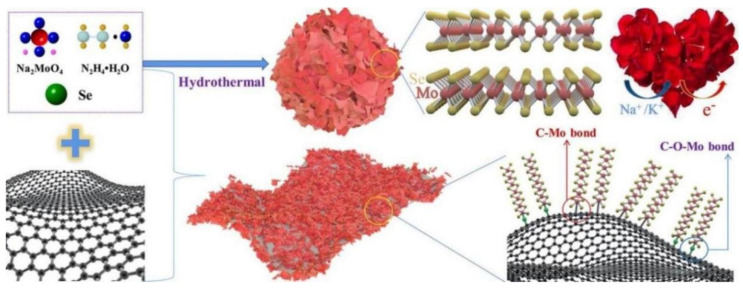
Schematic illustration of the synthetic procedure: MoSe_2_ nanosheets perpendicularly deposition onto the surface of rGO (MoSe_2_@rGO). Reproduced with permission [65]. Copyright 2021, American Chemical Society.

**Figure 5 nanomaterials-12-02837-f005:**
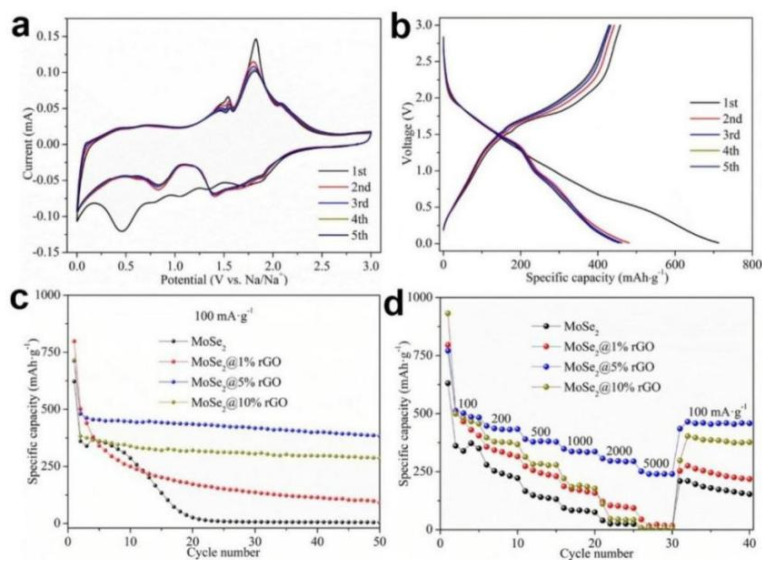
Electrochemical performance of MoSe_2_@rGO based SIBs: (**a**) CV and (**b**) GCD curves of MoSe_2_@5%rGO at 100 mA·g^−^^1^; (**c**) Cycling ability at the current density of 100 mA·g^−1^; (**d**) Rate capability from 100 to 5000 mA·g^−1^. Reproduced with permission [65]. Copyright 2020, Elsevier.

**Figure 6 nanomaterials-12-02837-f006:**
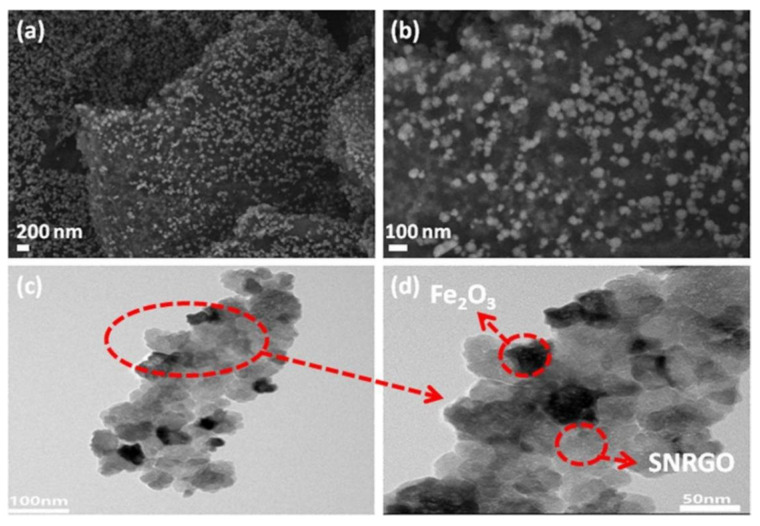
SEM images of (**a**) and (**b**) Fe_2_O_3_/SNRGO with different magnifications; TEM images of (**c**) and (**d**) Fe_2_O_3_/SNRGO with different magnifications. Reproduced with permission [56]. Copyright 2020, Elsevier.

**Figure 7 nanomaterials-12-02837-f007:**
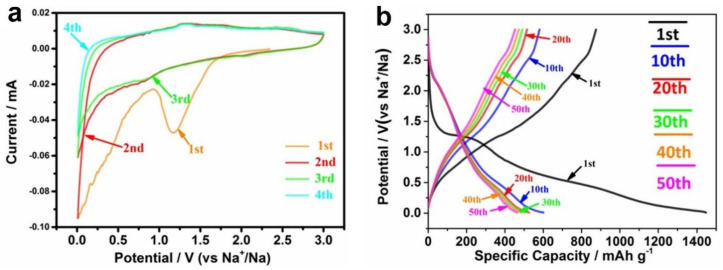
(**a**) CV of Fe_2_O_3_/SNRGO electrode, (**b**) GCD curves for the 1st, 10th, 20th, 30th, 40th and 50th cycles of Fe_2_O_3_/SNRGO electrode at 0.1 A g^−1^. Reproduced with permission [56]. Copyright 2020, Elsevier.

**Figure 8 nanomaterials-12-02837-f008:**
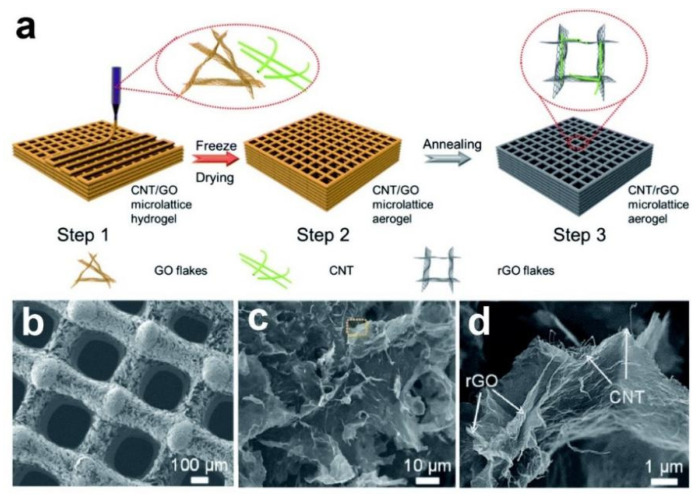
(**a**) The flow diagram illustrates the fabrication procedure of the printed three-dimensional rGO/CNT microlattice; (**b**–**d**) SEM images of rGO/CNT microlattice under different magnifications. Reproduced with permission [47]. Copyright 2020, Royal Society of Chemistry.

**Figure 9 nanomaterials-12-02837-f009:**
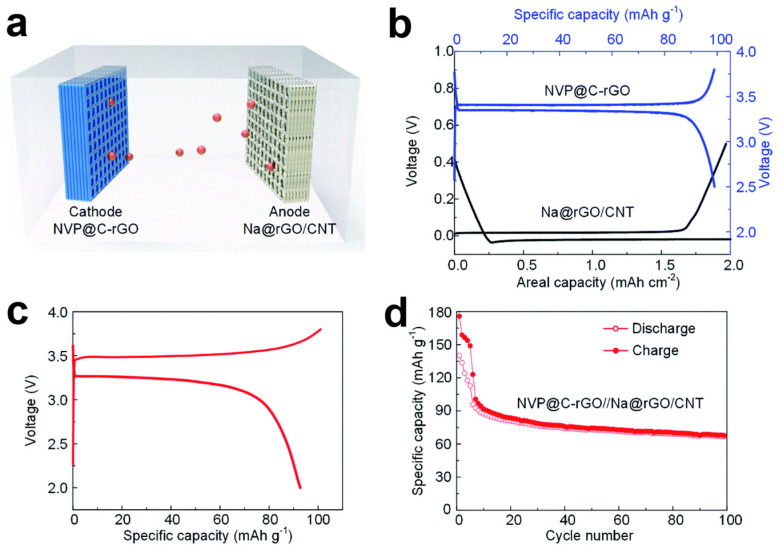
(**a**) Structure diagram of SIBs composite of cathode (NVP@C-rGO) and anode (Na@rGO/CNT); (**b**) GCD curves of the cathode (NVP@C-rGO, blue lines) and anode (Na@rGO/CNT, dark lines) half cells; (**c**) GCD curves of Na@rGO/CNT//NVP@C-rGO battery at 100 mA g^−1^; (**d**) The specific capacity change with the cycle number at the current density of 100 mA g^−^^1^. Reproduced with permission [47]. Copyright 2020, Royal Society of Chemistry.

**Figure 10 nanomaterials-12-02837-f010:**
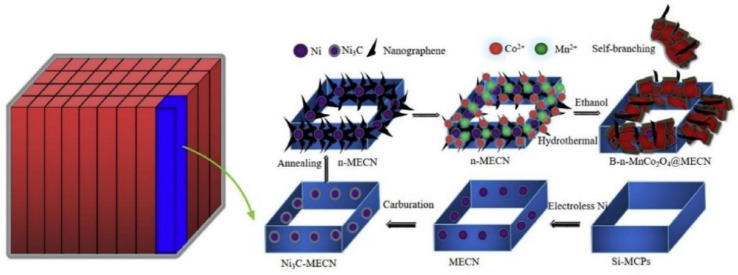
Schematic illustration of preparation steps of the MnCo_2_O_4_ deposit on the nanographene modified MECN for Na-ion batteries. Reproduced with permission [79]. Copyright 2020, Elsevier.

**Figure 11 nanomaterials-12-02837-f011:**
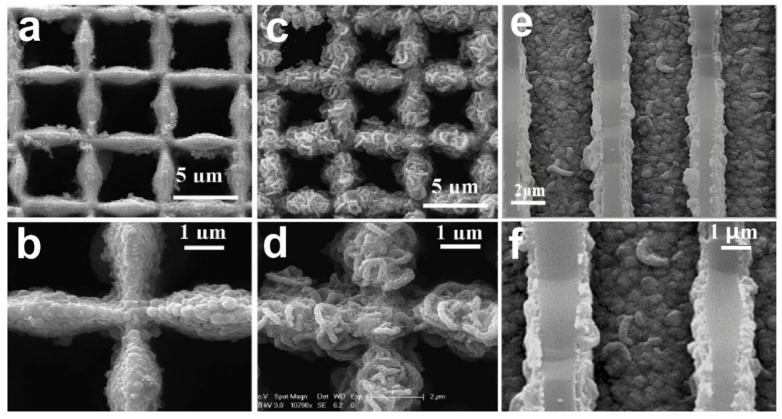
(**a**,**b**) Top view of n-MECN with different magnification; the SEM images of (**c**,**d**) Surface of MnCo_2_O_4_@n-MECN electrode with different magnification; (**e**,**f**) Cross-sectional images of MnCo_2_O_4_@n-MECN with different magnification. Reproduced with permission [79]. Copyright 2020, Elsevier.

**Figure 12 nanomaterials-12-02837-f012:**
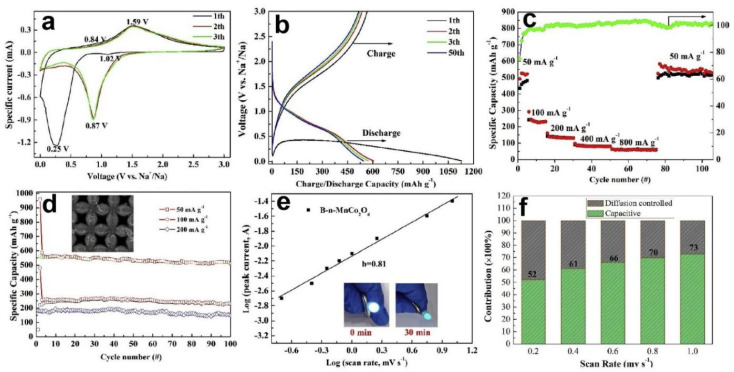
Electrochemical properties of the MnCo_2_O_4_@n-MECN based SIBs: (**a**) CV curves; (**b**) 1st, 2nd, 3rd, and 50th GCD curves at 50 mA g^−1^; (**c**) Rate capability performance and Coulombic efficiency from 50 mA g^−^^1^ to 800 mA g^−1^; (**d**) Cycling stability at 50, 100, and 200 mA g^−1^; (**e**) Relationship between log scanning rates and log cathodic peak currents; (**f**) Normalized contribution ratio at different scan rates. Reproduced with permission [79]. Copyright 2020, Elsevier.

**Figure 13 nanomaterials-12-02837-f013:**
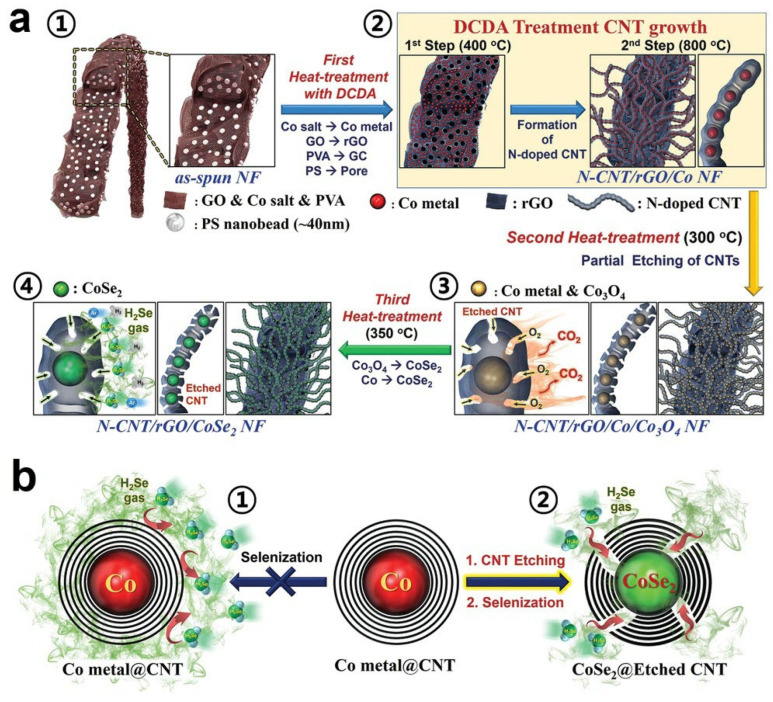
(**a**) Formation mechanism of the golden bristlegrass-like graphene matrixed nanofibers entangled with bamboo-like N-CNTs containing CoSe_2_ nanocrystals at each node and (**b**) efficient conversion of Co to CoSe_2_ by partial etching strategy of N-CNT wall. Reproduced with permission [90]. Copyright 2019, Wiley-VCH.

**Figure 14 nanomaterials-12-02837-f014:**
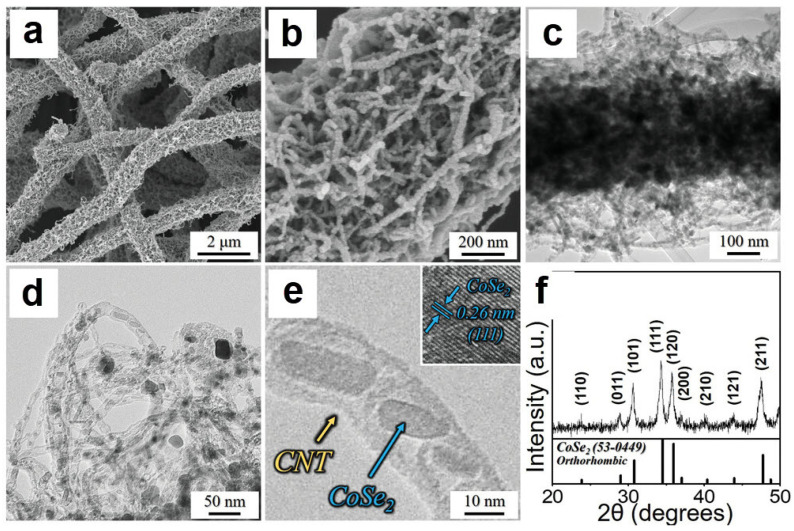
The SEM and TEM morphology and XRD characterization of N-CNT/rGO/CoSe_2_ NF: (**a**,**b**) FE-SEM images with different magnifications, (**c**–**e**) HR-TEM images with different magnifications, (**f**) XRD pattern. Reproduced with permission [90]. Copyright 2019, Wiley-VCH.

**Figure 15 nanomaterials-12-02837-f015:**
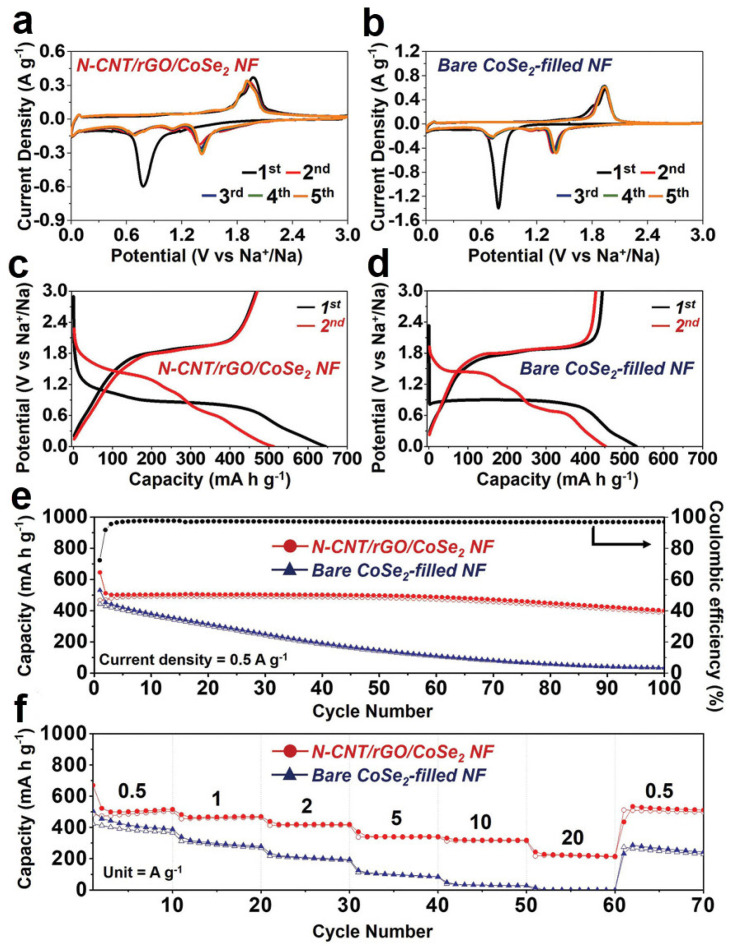
Electrochemical characteristics of N-CNT/rGO/CoSe_2_ NF and bare CoSe_2_-filled NF for SIBs assembled using carbonate-based electrolytes: (**a**,**b**) CV curves, (**c**,**d**) GCD curves at 0.5 A g^−1^, (**e**) Cycle ability at 0.5 A g^−1^, and (**f**) Rate properties. Reproduced with permission [90]. Copyright 2019, Wiley-VCH.

**Figure 16 nanomaterials-12-02837-f016:**
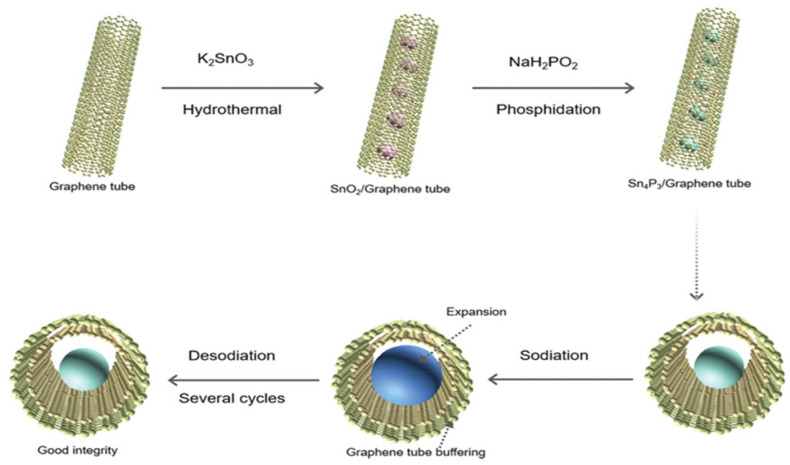
The fabrication steps the nanocomposite of Sn_4_P_3_/GT. The schematic illustration shows that the SnO_2_ nanoparticles were firstly prepared in the inside of the graphene tube by selective growth of hydrothermal method and then a phosphidation process was used to form Sn_4_P_3_/GT; the following three diagrams depict the sodiation/desodiation process of Sn_4_P_3_. Reproduced with permission [58]. Copyright 2021, Wiley-VCH.

**Figure 17 nanomaterials-12-02837-f017:**
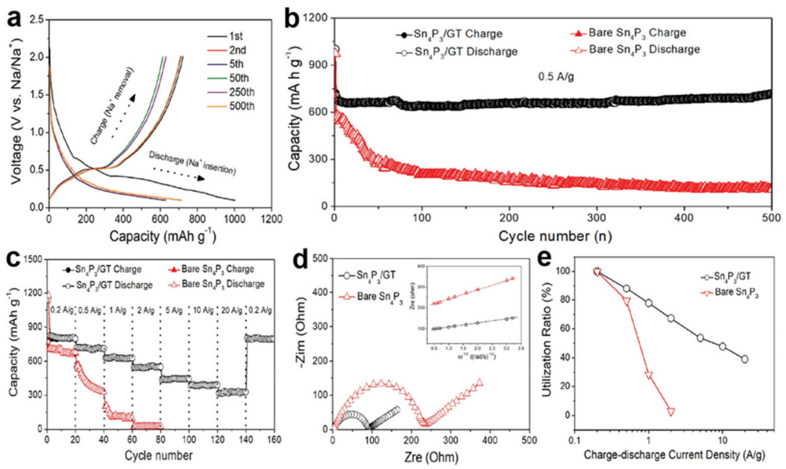
(**a**) GCD curves of Sn_4_P_3_/GT for the 1st, 2nd, 5th, 50th, 250th, and 500th cycles with the voltage window of 0.01–2 V versus Na/Na^+^ at 0.5 A g^−1^; the comparison of electrochemical properties between the electrode of Sn_4_P_3_/GT and bare Sn_4_P_3_: (**b**) GCD cycling tests at 0.5 A g^−1^, (**c**) rate capability under different current densities for 500 cycles, (**d**) Nyquist plot, Inset of (**d**) shows the relationship between ω^–1/2^ (where ω is the angular frequency in the low-frequency region, ω = 2πf) and the real part of the impedance spectra (Z_re_), (**e**) Utilization radio of the active material at different GCD current densities. Reproduced with permission [58]. Copyright 2021, Wiley-VCH.

**Figure 18 nanomaterials-12-02837-f018:**
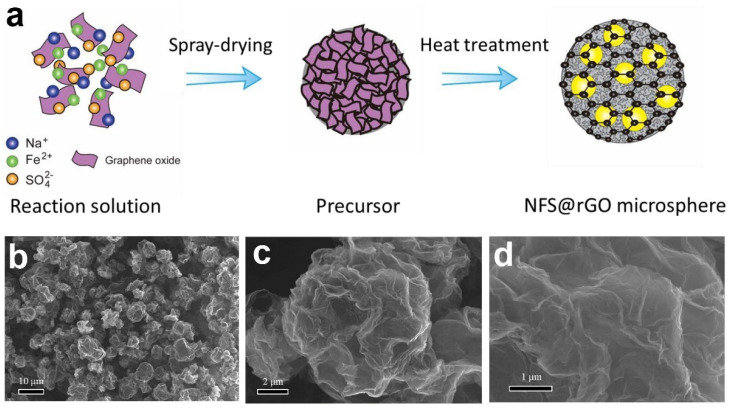
(**a**) The fabrication steps of NFS@rGO microspheres; (**b**–**d**) SEM images of graphene coated Na_2_._4_Fe_1_._8_(SO_4_)_3_ microspheres with different magnifications. Reproduced with permission [99]. Copyright 2021, Elsevier.

**Figure 19 nanomaterials-12-02837-f019:**
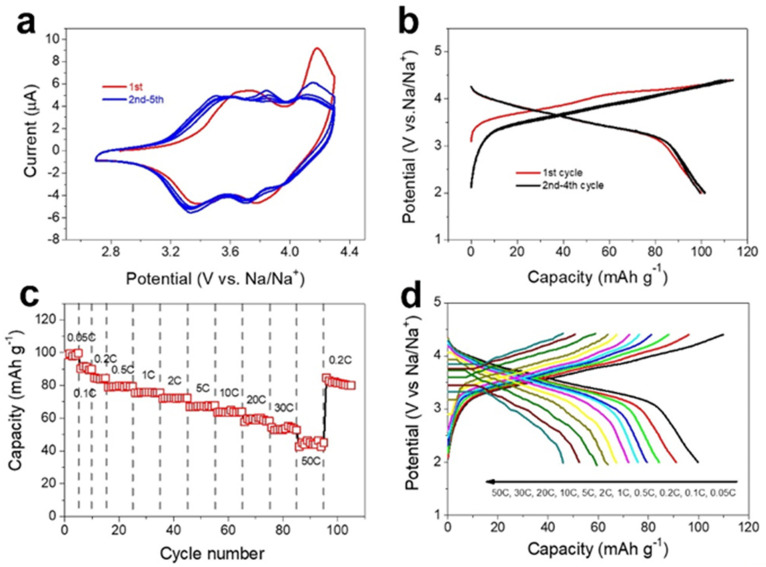
Electrochemical properties of the NFS@rGO microspheres-based electrode: (**a**) CV curves at a scan rate of 0.1 mV s^−1^, (**b**) GCD curves performed at 0.05 C (1 C = 100 mA g^−1^), (**c**) Rate capacity performance, (**d**) GCD curves at different current density. Reproduced with permission [99]. Copyright 2021, Elsevier.

**Figure 20 nanomaterials-12-02837-f020:**
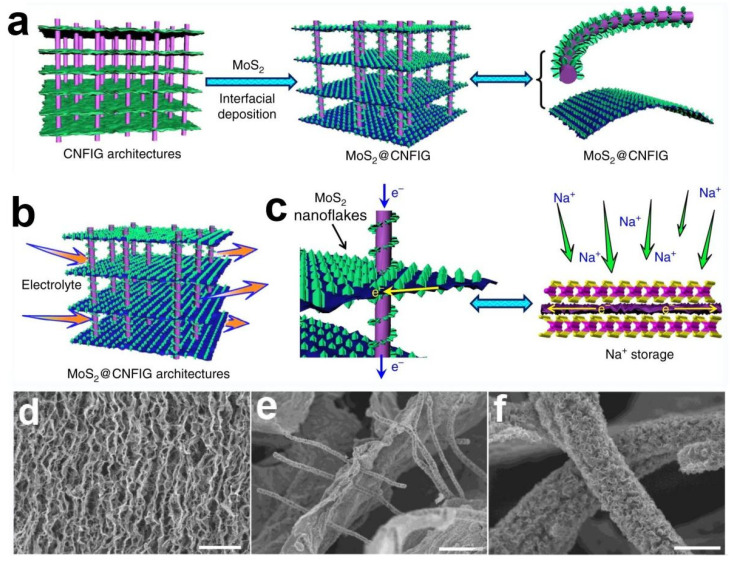
Schematic illustration of (**a**) homogeneous deposition of MoS_2_ nanoflakes on layered CNFIG matrix, (**b**) The mechanism of porous morphology of MoS_2_@CNFIG nanocomposite accelerates the penetration of electrolyte and (**c**) the mechanism of electrically conductive MoS_2_@CNFIG facilitates the quick sodiation/desodiation ability and rapid transfer of sodium ions; SEM images of the MoS_2_@CNFIG nanocomposite: (**d**) overall structure with a scale bar of 50 μm and (**e**) cross section with scale bars of 5 μm; (**f**) MoS_2_ layers uniformly deposition on the graphene layer and CNF bridges with a scale bar of 2 μm. Reproduced with permission [118]. Copyright 2019, Springer Nature.

**Figure 21 nanomaterials-12-02837-f021:**
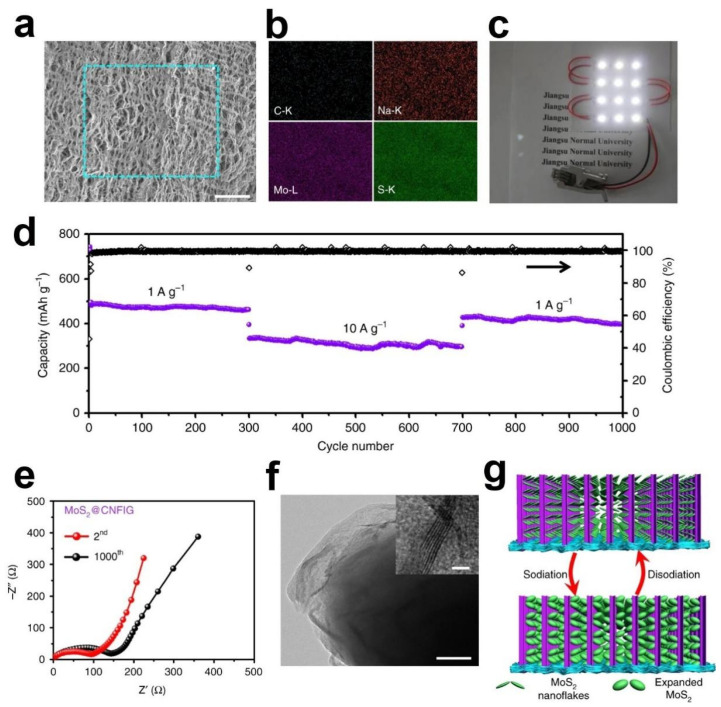
(**a**) SEM image (scale bar = 50 μm) and (**b**) corresponding EDX elemental mappings (C, Na, Mo, and S) of MoS_2_@CNFIG anode after 1000 cycles; (**c**) Three assembled full cells based on MoS_2_@CNFIG anodes in parallel powering the 12 LEDs; (**d**) Long-term cycling ability and Coulomb efficiency of the full cell at 1 and 10 A g^−1^; (**e**) Nyquist plots of the full cell after 2 and 1000 cycles; (**f**) TEM image of the full cell after 1000 cycles with the scale bar of 50 nm and 2 nm (inset in (**f**)); (**g**) Schematic illustration of porous space of CNFIG provided the volume expansion space for MoS_2_ nanoflakes. Reproduced with permission [118]. Copyright 2019, Springer Nature.

**Figure 22 nanomaterials-12-02837-f022:**
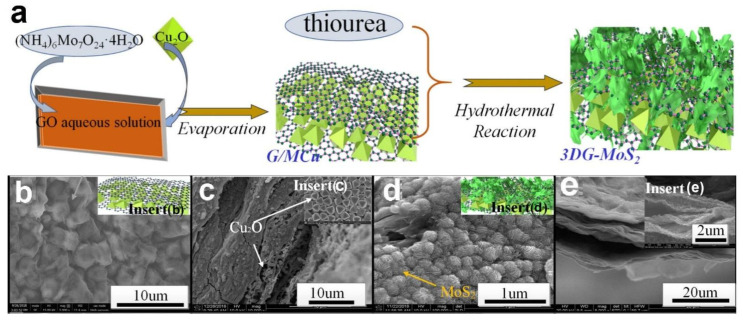
(**a**) Schematic illustration of the fabrication processes of 3DG-MoS2; (**b**) FE-SEM images of section of G/MCu and schematic structure of G/MCu (insert of (**b**)); (**c**) Cross-section FE-SEM images of G/MCu and further amplification of c (Insert of (**c**)); (**d**) FE-SEM images of 3DG-MoS_2_ surface and schematic structure of 3DG-MoS_2_ (Insert (**d**)); (**e**) Cross-section FE-SEM images of 3DG-MoS_2_ with different magnifications. Reproduced with permission [30]. Copyright 2020, John Wiley and Sons.

**Figure 23 nanomaterials-12-02837-f023:**
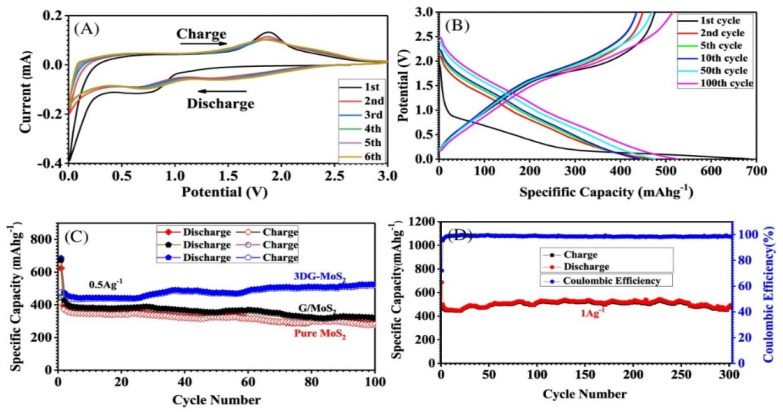
(**A**) CV curves of the 3DG-MoS_2_ electrode at the voltage range of 0.0 to 3.0 V versus Na/Na^+^ under the scan rate of 0.5 mVs^−1^; (**B**) GCD curves of the 3DG-MoS_2_ electrode at 0.5 A g^−1^ under different cycles; (**C**) Charge/discharge capacities comparison among 3DG-MoS_2_, MoS_2_ nanosheets and G/MoS_2_ at 0.5 A g^−1^; (**D**) Charge/discharge capacities and Coulomb efficiency of the 3DG-MoS_2_ electrode at 1 A g^−1^. Reproduced with permission [30]. Copyright 2020, John Wiley and Sons.

## Data Availability

Not applicable.

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
