# Peer review of "Recent Progress on Graphene-Based Nanocomposites for Electrochemical Sodium-Ion Storage"

_nanomaterials, 2022, doi:10.3390/nano12162837_

Round 1
Reviewer 1 Report
This a nice and comprehensive review on the use of graphene based structure in sodium ion batteries. I recommend its publication in Nanomaterials. I only have a few suggestions for improvements:
1.) On line 38: "Therefore sodium ion batteries have a ... high half-cell potential (0.3 V higher than lithium-ion battery)" : This statement appears to be vague as it creates the impression as if Na anodes would be more energetic than Li ones. A less misunderstandable formulation of the same would be that the standard electrode potential of the Na metal anode is only 0.3 V less negative than that of the Li metal anode therefore Na ion cells are expected to be similarly energetic as Li ones.
2.) "sodiation/desodiation of Na ions" line 117. Probably intended to be "sodiation/desodiation of the cathode active species".
3.) The acronym SNRGO is not defined anywhere. I assume it means sulfur and nitrogen co-doped rGO. Another undefined acronym is NSF, first occurring in the caption of Fig 17. All the acronyms in the manuscript should be properly defined.
Author Response
Dear reviewer,
Thank you for your comments concerning our manuscript entitled “Recent progress on Graphene-based Nanocomposites for Elec-trochemical Sodium-Ion Storage” (ID: nanomaterials-1849405). Those comments are all valuable and helpful for revising and improving our paper, as well as the important guiding significance to our researches.
We have studied the comments carefully and have made adjustments and corrections to meet the requirements of experts with positive responses.
All the revisions to the manuscript have been marked up using the “Track Changes” function.
We would love to thank you for allowing us to resubmit a revised copy of the manuscript and we highly appreciate your time and consideration.
Best regards
Sincerely yours
The main corrections in the paper and the responses to the reviewer’s comments are as follows:
Response to comment: On line 38: "Therefore sodium ion batteries have a ... high half-cell potential (0.3 V higher than lithium-ion battery)": This statement appears to be vague as it creates the impression as if Na anodes would be more energetic than Li ones. A less misunderstandable formulation of the same would be that the standard electrode potential of the Na metal anode is only 0.3 V less negative than that of the Li metal anode therefore Na ion cells are expected to be similarly energetic as Li ones.
1. Response: We are grateful for these suggestions. We appreciate the comments by Reviewer #1. It was our negligence that didn't make it clear. The electrochemical potential of sodium-ion (-2.71V, Na+/Na) is only 0.3V smaller than that of lithium ion (-3.04V, Li+/Li), both are expected to have similarly energetic. We have adjusted the contend.
2. Response to comment: "sodiation/desodiation of Na ions" line 117. Probably intended to be "sodiation/desodiation of the cathode active species".
Response: Thank you for your work on the careful identified the problems of our article. We are very sorry for our negligence of this mistake before upload the article. We have already revised it.
3. Response to comment: The acronym SNRGO is not defined anywhere. I assume it means sulfur and nitrogen co-doped rGO. Another undefined acronym is NSF, first occurring in the caption of Fig 17. All the acronyms in the manuscript should be properly defined.
Response: Thank you for your work on the careful identified the problems of our article. It’s really a good question and we add the correct definition of SNRGO and NSF in our artical.
Special thanks to you for your good comments.

Reviewer 2 Report
Sodium-ion batteries (SIBs analog to Li-ion) have emerged as a promising alternative to Lithium-ion batteries (LIBs) for energy storage applications, due to abundant sodium resources, low cost, and similar electrochemical performance. Electrical energy storage plays a vital role in consumer electronics and transportation systems. Among the available energy storage system, Electrochemical energy storage devices such as rechargeable batteries require high energy density with rapid charge-discharge rates. This can be met only with the next generation material. Therefore, the chosen area and topic by author Z. Meng et al are significant to the chosen journal Nanomaterials. The authors have reviewed recent progress in designing graphene-based nanocomposite anodes for SIBs, with a focus on preparation, ion transportation, and its challenges.
The review is well organized with a comprehensive summary of the recent updates in the graphene-related electrodes for SIBs. However, some revision is required before rendering a final decision.
My specific points are below:
· A brief overview of the SIB with a schematic illustration of the working mechanism can be included for clarity to readers.
· The working principle of LIB to contrast with SIBs can be discussed based on the literature (doi.org/10.1016/j.progsolidstchem.2020.100298; and doi.org/10.3390/en13061477).
· A broad overview of the synthesis of graphene, GO/graphite oxide, and its reduction using various methods for novel applications in SIBs. Can be included.
· Are the conventional carbonate-based electrolytes suitable for SIBs?
· How does graphene differs from carbon structure as a “house of cards” model? How graphene has been traditionally produced as pristine and with dopants?
· Maybe a good idea to classify the graphene anode materials as intercalation (insertion) based, conversion based, and alloying-based materials.
· Page 3, in the first paragraph lines 88-93 “hinders the overall performance of SIBs”, arguably materials such as MnO2, EMD, NaNiPO4, NaMn1/3Ni1/3Co1/3PO4 have been shown to be feasible for sodium-ion devices reported by Manickam Minakshi et al. Please discuss.
· Page 4, line 117 “And the synergistic…” this sentence is unclear.
· Page 4, line 132 – is it figure 1? Should be stated as Figure 2.
· Page 5, line 167: should read as S K Chong et al….
· Page 6, results are provided but need some reasoning.
· Page 8, first paragraph – what is the message additive-free composite is good or bad?
· Page 9, line 293 A g-1.
· Page 10, line 303 “As described in previous report” reference, please.
· Page 12, line 354 – “capacitive contribution” where it comes from graphene?
· Page 13, line 395 – Figure 12a-4. The word “a” is missing. The explanation for Figure 12b is missing.
· Incorporation of graphene into Fe2O3 nanostructures has reported being greatly improved the electrochemical performance of the electrode with satisfactory sodium storage properties due to the synergistic effect. However, how the hydrothermal approach enhances the storage properties?
· Why the heteroatom doping enhances the storage in graphene nanocomposite anodes?
· Section 2.3 what is the rationale for encapsulation of active materials in graphene? Is it to enhance electron transfer reaction kinetics and its storage activity? If so, what is the role of N dopants/core-shell structure?
· Section 2.4; the sodiation/desodiation ability of sodium ions and its conducting pathway are well explained in the literature for SIBS, please cite (doi.org/10.1016/j.mtener.2018.08.004; 10.1039/C8NR03824D) and discuss.
· Page 22, line 647: “Form” should read as “From”
· Figure 21, text related – please explain the role of thiourea in hydrothermal synthesis.
· Page 23; line 673 “penetration of electrolyte” what is the electrolyte used?
· Page 24; line 682 “the author investigated” reference, please.
· How to prevent the formation of the SEI layer?
· Please provide future perspectives in this space.
Author Response
Dear reviewer,
Thank you for your comments concerning our manuscript entitled “Recent progress on Graphene-based Nanocomposites for Elec-trochemical Sodium-Ion Storage” (ID: nanomaterials-1849405). Those comments are all valuable and helpful for revising and improving our paper, as well as the important guiding significance to our research.
We have studied the comments carefully and have made adjustments and corrections to meet the requirements of experts with positive responses.
All the revisions to the manuscript have been marked up using the “Track Changes” function.
We would love to thank you for allowing us to resubmit a revised copy of the manuscript and we highly appreciate your time and consideration.
The main corrections in the paper and the responses to the reviewer’s comments are list in the word file.
Best regards
Mai Li

Round 2
Reviewer 2 Report
In this reviewer’s opinion, the revised version and the highlighted parts are OK. In which, the authors have taken my queries into a consideration and revised the manuscript.